# Measurement report: In situ observations of deep convection without lightning during the tropical cyclone Florence 2018.

Clara M. Nussbaumer[1], Ivan Tadic[1], Dirk Dienhart[1], Nijing Wang[1], Achim Edtbauer[1], Lisa Ernle[1], Jonathan Williams[1,2], Florian Obersteiner[3], Isidoro Gutiérrez-Álvarez[4], Hartwig Harder[1], Jos Lelieveld[1,2], and Horst Fischer[1]

[1]Max Planck Institute for Chemistry, Department of Atmospheric Chemistry, Mainz, Germany
[2]Climate and Atmosphere Research Center, The Cyprus Institute, Nicosia, Cyprus
[3]Karlsruhe Institute of Technology, Karlsruhe, Germany
[4]Department of Integrated Sciences, Center for Natural Resources, Health and Environment (RENSMA), University of Huelva, Spain

**Correspondence:** Clara M. Nussbaumer (clara.nussbaumer@mpic.de)

**Abstract.** Hurricane Florence was the sixth named storm in the Atlantic hurricane season 2018. It caused dozens of deaths and major economic damage. In this study, we present in situ observations of trace gases within tropical storm Florence on September 2, 2018 after it had developed a rotating nature, and of a tropical wave observed close to the African continent on August 29, 2018 as part of the research campaign CAFE Africa (Chemistry of the Atmosphere - Field Experiment in Africa) with the HALO (High Altitude Long Range) research aircraft. We show the impact of deep convection on atmospheric composition by measurements of the trace gases nitric oxide (NO), ozone ($O_3$), carbon monoxide (CO), hydrogen peroxide ($H_2O_2$), dimethyl sulfide (DMS) and methyl iodide ($CH_3I$), and by the help of color enhanced infrared satellite imagery taken by GOES-16. While both systems, the tropical wave and the tropical storm, are deeply convective, we only find evidence for lightning in the tropical wave using both in situ NO measurements and data from the World Wide Lightning Location Network (WWLLN).

## 1 Introduction

Tropical cyclones are low-pressure systems evolving over warm tropical waters usually close to the equator ($\pm\,20°$) - an area which includes the so-called Intertropical Convergence Zone (ITCZ) (Frank and Roundy, 2006; Deutscher Wetterdienst). The ITCZ is a global band of convection where south- and northeasterly trade winds converge. It is characterized by rapidly changing weather events. Air heated by the sun near the equator rises, creating low pressures near the surface, which initiates flows from adjacent areas (Waliser and Gautier, 1993; Wang and Magnusdottir, 2006; Deutscher Wetterdienst). In this region of high ocean temperature and intense solar radiation humid air can rise deeply into the troposphere up to 15 km and higher (Collier and Hughes, 2011; Deutscher Wetterdienst). This is associated with the formation of deep, convective cumulonimbus clouds accompanied by heavy rainfall and thunderstorm activity (Zipser, 1994; Xu and Zipser, 2012). In the early stages, these systems are referred to as tropical waves or disturbances which together with low wind shear and high ocean temperature of 26.5 °C or higher can form tropical cyclones (Frank and Roundy, 2006; Shapiro and Goldenberg, 1998; National Ocean

Service, 2020b). Wu et al. suggested that simultaneous occurring of convection and vorticity in disturbances favors tropical cyclone formation. However, the exact formation mechanism of tropical cyclones from tropical waves is not fully understood today (Wu and Takahashi, 2018; Frank and Roundy, 2006). Tropical cyclones are characterized by their rotating nature around a center originating from Coriolis forces and the balance of the pressure gradient (Smith et al., 2005; Gray, 1975). Consequently, tropical cyclones in the northern hemisphere spin counter-clockwise and in the southern hemisphere clockwise, while rotating systems do not develop within about 5°of the equator (National Ocean Service, 2020a; Gray, 1975). Tropical cyclones are categorized and named according to the maximum sustained wind speed and their geographic origin. A tropical cyclone formed over the Atlantic Ocean - most often west of the African continent - with a maximum wind speed of 64 knots (118 km/h) and higher is defined as a hurricane according to the Beaufort scale. Lower wind speeds of up to 34 knots (63 km/h) characterize a tropical depression. Tropical cyclones with intermediate wind speed (34 to 63 knots) are referred to as tropical storms (DeMaria et al., 2012; National Weather Service).

Deep convection can affect trace gas concentrations in the upper troposphere which was for example shown by Dickerson et al. (1987) who reported increased concentrations of NO, CO, $O_3$ and other reactive species in a thunderstorm outflow over the Midwestern United Stated in 1985 and Barth et al. (2015), the latter based on observations during the DC3 (Deep Convective Clouds and Chemistry) field campaign. Similar observations were made by Bucci et al. (2020) who reported convective uplift in the upper troposphere / lower stratosphere based on the StratoClim aircraft campaign over Nepal in 2017. It is usually observed along with lightning. Collision of light ice particles moving upwards in cumulonimbus clouds and graupel particles moving downwards due to gravity in the presence of supercooled water induces electric charge separation which accumulates and discharges spontaneously as a lightning flash (Lal et al., 2014; Liu et al., 2012). Lightning events are frequent over tropical continental areas such as South America or Africa where cloud-to-cloud lightning contributes by around 90 % (Williams and Sátori, 2004; Price and Rind, 1993). While lightning events over the ocean are less frequent, they are subject to extensive research with regards to the occurrence in tropical cyclones. Zipser (1994) reported significantly reduced lightning activity over tropical oceans despite heavy rainfall from convective clouds in comparison to tropical continental and coastal areas with similar rainfall based on shipborn observations during the Global Atmospheric Research Program Atlantic Tropical Experiment in 1974 (GATE). Lal et al. observed higher lightning activity over continental compared to oceanic areas based on satellite data from 2000 to 2011 (Lal et al., 2014). These results are in line with other published works e.g. Cecil et al. (2014), Xu and Zipser (2012) and Xu et al. (2010). Regarding tropical cyclones, DeMaria et al. (2012) reported more intense lightning activity in tropical storms compared to hurricanes based on lightning data from the World Wide Lightning Location Network (WWLLN) and satellite data on tropical cyclones which is in agreement with findings from Abarca et al. (2011) who reported a decrease in lightning density with increasing storm strength. DeMaria et al. additionally found greater lightning activity in intensifying compared to weakening storms which was also reported by Zhang et al. (2015).

Measurements of nitric oxide (NO) in the upper troposphere can provide indirect evidence on the recent occurrence of lightning. The heat developed in lightning flashes allows the abundant $N_2$ and $O_2$ to atomize and then recombine to form NO (Murray, 2016; Huntrieser et al., 2011). In the upper troposphere, lightning is the main source of NO by around 80 % whereas it only contributes about 10 % to the overall global NO budget (Schumann and Huntrieser, 2007; Murray, 2016). Over

the ocean, the only significant NO emissions are from lightning, ships and aircraft (Bond et al., 2002; Masiol and Harrison, 2014). NO sources over land are more versatile including anthropogenic emissions from industry, vehicles and biomass burning (partly natural) as well as natural sources from lightning and soil, the latter dominating over West Africa (Pacifico et al., 2019; Knippertz et al., 2015). One lightning flash produces approximately $2\text{-}40 \times 10^{25}$ molecules of NO, which together with $NO_2$ as $NO_x$ have a lifetime of several days near the equator (Pollack et al., 2016; Schumann and Huntrieser, 2007; Levy et al., 1999).

Other trace gases can be used to detect convective injection from the marine boundary layer into the upper troposphere. These include near-surface emissions of carbon monoxide (CO) from the photolysis of dissolved organic matter (DOM) (Stubbins et al., 2006), and methyl iodide ($CH_3I$) which is produced by algae and phytoplankton as well as aqueous photochemical processes and is released from the ocean with an atmospheric lifetime of 4-7 days (Tegtmeier et al., 2013; Bell et al., 2002). Another possible source could be dust originated from the African continent which enters the sea or gets in contact with sea water vapor and produces methyl iodide as described by Williams et al. (2007). One possible explanation for the formation of $CH_3I$ is a substitution reaction of methoxy group containing species and iodide from seawater under the presence of iron ions from dust. However, the mechanism is not yet finally understood (Williams et al., 2007). Furthermore, phytoplankton forms dimethylsulphoniopropionate (DMSP) in seawater which is degraded to dimethyl sulfide (DMS) and emitted from the ocean's surface (Simó and Dachs, 2002; Gondwe et al., 2003). Its lifetime ranges from 1 to 2 days and depends on the atmospheric abundance of OH and $NO_3$ which oxidize DMS (Breider et al., 2010). OH concentrations in turn are controlled by nitrogen oxides $NO_x$ (= NO + $NO_2$) and ozone. The latter is formed by photolysis of $NO_2$ with $O_2$ and depends on ambient NO and hydrocarbons (Nussbaumer and Cohen, 2020). Photolysis of $O_3$ and reaction with water vapor yield OH (Levy, 1971; Lelieveld and Dentener, 2000; Tegtmeier et al., 2013). $NO_x$ from lightning plays a key role in OH formation in the free troposphere (Lelieveld et al., 2018; Brune et al., 2018). On the other hand, in $NO_x$-poor conditions, e.g. in the marine boundary layer, close to the surface, the concentrations of peroxyradicals can build up, leading to $O_3$ destruction and high levels of $H_2O_2$ (Ayers et al., 1992). Due to a strong surface uptake loss of $H_2O_2$, concentrations peak at mid-range altitudes (Weller and Schrems, 1993).

Hurricane Florence was the sixth named storm in the Atlantic hurricane season of 2018 (AL062018) (Stewart and Berg, 2019). It developed from a tropical wave over the West African continent which was first reported on August 28 by the National Hurricane Center (NHC) Miami (FL, USA) (National Hurricane Center, 2018c). On August 31, a tropical depression developed which was upgraded to being a tropical storm by the NHC on September 1 (National Hurricane Center, 2018a; Stewart and Berg, 2019). The tropical cyclone grew to hurricane strength on September 4 (National Hurricane Center, 2018b). Hurricane Florence reached its maximum wind speed of 130 knots (category 4 hurricane) on September 11 and made landfall on September 14 in North Carolina. It claimed overall more than 50 deaths and caused 24 billion US dollars worth of damage mainly from floodings in North and South Carolina (Stewart and Berg, 2019; Paul et al., 2019).

Some studies have investigated trace gas concentrations and convective uplift in the upper troposphere through aircraft observations. Newell et al. (1996) reported in-situ observations of deep convection in the Typhoon Mireille in 1991 which they found to be strongest in the wall cloud region. They additionally detected enhanced NO concentrations in the eye wall

area and suggested lightning as a source based on observations reported by Davis et al. (1996). Roux et al. (2020) found the convective uplift of boundary layer air as well as the inflow of lower stratospheric air to the upper troposphere based on measurements of CO, $O_3$ and $H_2O$ during aircraft typhoon observations over Taipei in 2016. In contrast, studies of lightning

activity within convective systems over the ocean and in tropical cyclones are predominantly based on satellite data and ground-based observations from the WWLLN (University of Washington; Abreu et al., 2010; Bürgesser, 2017; Hutchins et al., 2012b; Bucsela et al., 2019) Generally, data from in situ chemical measurements in the upper troposphere are sparse and to our knowledge, the in situ aircraft observation of deep convection in tropical cyclones accompanied by and in the absence of lightning depending on the stage of development has not been reported before. In this study, we present airborne in situ

observations of trace gases within a tropical wave on August 29, 2018 and of the tropical storm Florence on September 2, 2018 based on measurements during the aircraft campaign CAFE Africa (Chemistry of the Atmosphere: Field Experiment in Africa). The data are examined for evidence of the chemical impacts of deep convection and lightning activity.

## 2  Observations

The research campaign CAFE Africa took place from August to September 2018 over the West African continent and the

105 central eastern Atlantic. 14 measurement flights (MF) were performed with the HALO (High Altitude Long Range) research aircraft starting from the campaign base in Sal on Cape Verde (16.75 °N, 22.95 °W). A detailed description of the campaign is provided by Tadic et al. (2021). In this paper, we report observations based on three measurement flights - MF10, MF12 and MF14. Figure 1 shows an overview of the geographical locations of the three flights including satellite images obtained from NASA Worldview on the day of observation. The images were taken by the VIIRS (Visible Infrared Imaging Radiometer Suite)

instrument carried by the NASA/NOAA satellite Suomi NPP (National Polar orbiting Partnership) based on a daily resolution (NASA Worldview, 2020). MF10 was carried out on August 24, 2018 and was chosen as comparison flight as parts of it were in close geographical proximity to MF14. We have restricted our analysis to data from MF10 which were obtained in these parts in a similar geographical area and altitude range as MF14 (as shown in Figure 1a). MF12 was carried out on August 29, 2018 and overpassed a tropical low-pressure system which had recently moved off the West African coast. The tropical cyclone

Florence was overpassed on September 2, 2018 in the scope of MF14 west of the Cape Verde islands after it was upgraded to being a tropical storm.

The research aircraft carried multiple instruments for the measurement of various atmospheric trace gases including NO, $O_3$, CO, $H_2O_2$, $CH_3I$ and DMS. NO was measured via chemiluminescence (detector from ECO Physics CLD 790 SR, Dürnten, Switzerland) with a relative uncertainty of 6 % and a detection limit of 5 $ppt_v$ (Tadic et al., 2020). $O_3$ mixing ratios were ana-

120 lyzed by UV absorption and chemiluminescence with the FAIRO instrument (chemiluminesence data with a total uncertainty of 2.5 %, Zahn et al. (2012)). CO was measured via quantum cascade laser absorption spectroscopy with an uncertainty of 4.3 % (Schiller et al., 2008). $H_2O_2$ mixing ratios were measured via dual-enzyme detection (modified AEROLASER AL2021, Garmisch-Partenkirchen, Germany) with a total measurement uncertainty of 9 % and a detection limit of 15 $ppt_v$ (Hottmann et al., 2020). DMS was measured via proton-transfer-reaction time-of-flight mass spectrometry (PTR-TOF-MS-8000, Ionicon

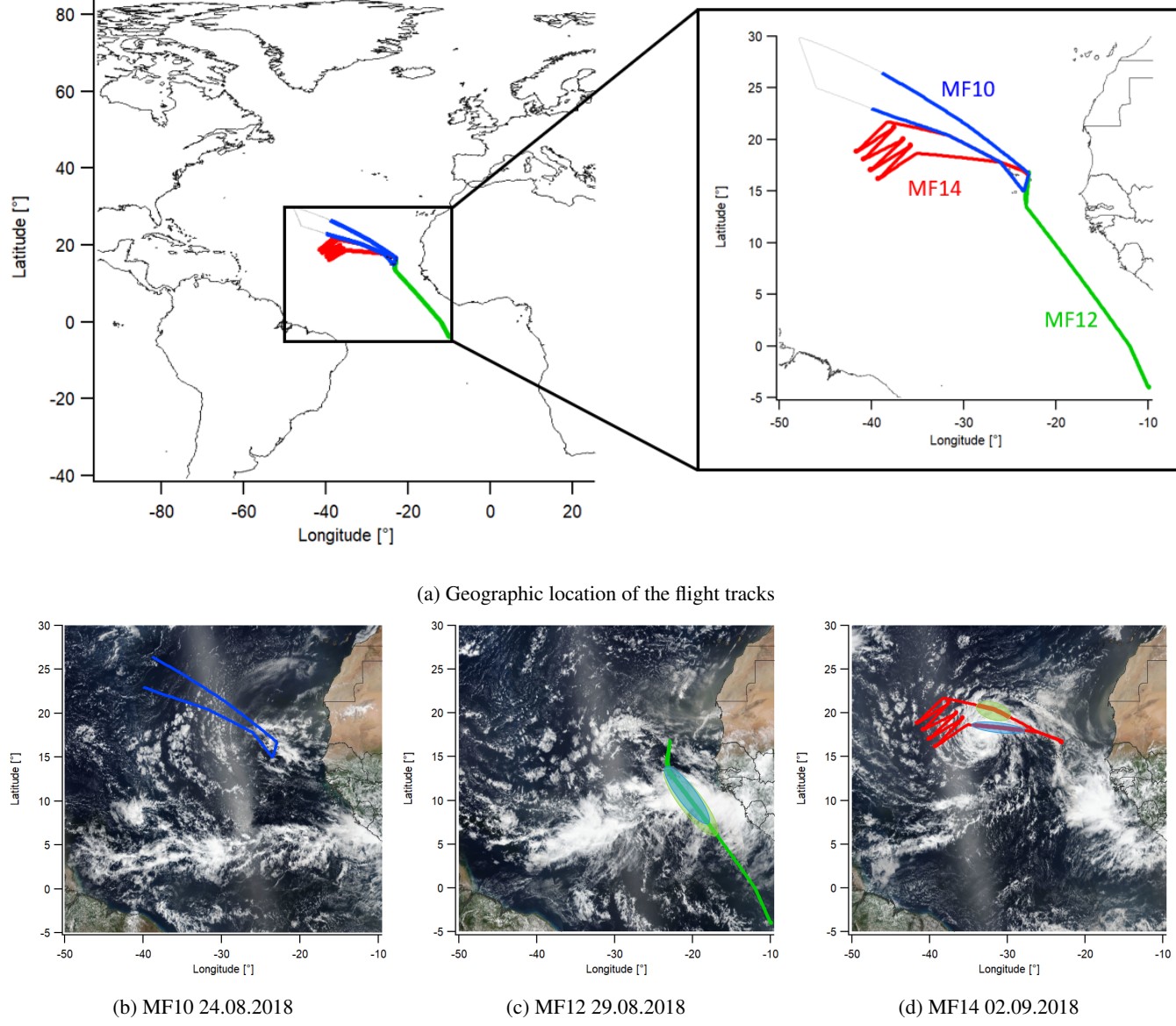

(a) Geographic location of the flight tracks

(b) MF10 24.08.2018      (c) MF12 29.08.2018      (d) MF14 02.09.2018

**Figure 1.** Overview of the flight tracks including satellite images obtained on the day of observation from the NASA Worldview application (NASA Worldview, 2020). Blue: MF10 on August 24 as comparison flight. Green MF12 on August 29 over the tropical wave. Red: MF14 on September 2 over the tropical storm Florence. Marked areas indicate convection.

Analytik GmbH, Innsbruck, Austria) with a detection limit of $15\,\mathrm{ppt}_v$ (Wang et al., 2020; Edtbauer et al., 2020). $CH_3I$ was measured with a custom-built fast gas chromatography - mass spectrometry system described by Bourtsoukidis et al. (2017) with a detection limit of $0.5\,\mathrm{ppt}_v$. Backward trajectories were calculated using the Lagrangian particle dispersion model FLEX-

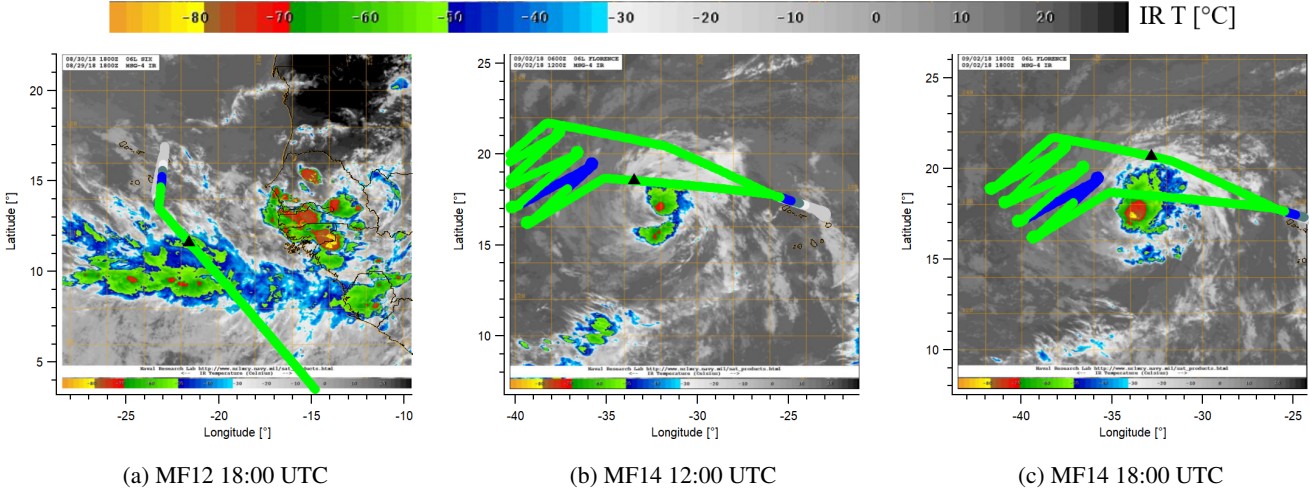

| (a) MF12 18:00 UTC | (b) MF14 12:00 UTC | (c) MF14 18:00 UTC |

**Figure 2.** Flight tracks with color enhanced infrared imagery obtained from the Naval Research Laboratory Tropical Cyclone page (Naval Research Laboratory, 2020) for MF12 and MF14 (no image availability before MF12 UTC 18:00). The altitude of the flight tracks is colored according to the IR temperature legend. The temperature during most parts of the flight was between -50 and -60 °C. Black triangles show the according position of the research aircraft for the time of the background IR image.

PART 10.2 (Stohl et al., 2005; Pisso et al., 2019). Lightning data were obtained from the WWLLN (University of Washington). Satellite images were acquired from NASA Worldview (2020). Color Enhanced Infrared Imagery were obtained from the Naval

Research Laboratory and from the Tropical Cyclone Realtime web page maintained by the Cooperative Institute for Research in the Atmosphere (CIRA), Colorado State University, and NOAA's Center for Satellite Research, Fort Collins Colorado (Naval Research Laboratory, 2020; CIRA and NOAA, 2018).

## 3 Results and Discussion

### 3.1 Cloud Top

Figure 2 shows the color enhanced infrared imagery obtained from the Naval Research Laboratory including the flight tracks for MF12 and MF14. The satellite images are colored according to the temperature deduced from IR emissions of cloud tops in °C as measured by the satellite GEOS-16 (Geostationary Operational Environmental Satellite). The flight track is colored according to the IR temperature scale showing the ambient temperature measured on the research aircraft which was mainly between -50 and -60 °C. The IR images give information on the occurrence of convective clouds. It can be assumed that the IR

temperature of a cloud top is equal to the ambient temperature at that altitude. Accordingly, lower IR temperatures represent clouds at higher altitudes. The flight altitude for MF12 at 18:00 UTC shown in Figure 2a was 14.4 km while overpassing an area of elevated clouds. The according temperature was -68 ± 1 °C, so the aircraft was likely above the cloud top. For MF12, the same area was passed in the morning at an altitude of 12.9 km, but no IR image is available. The temperature was

-56 $\pm$ 1 °C. Assuming a similar cloud elevation in the morning, the research aircraft was at a similar altitude as the cloud top. The flight altitude for MF14 at 12:00 UTC and at 18:00 UTC as shown in Figures 2b and 2c was 13.2 km and the temperature was -58 $\pm$ 1 °C. The colored IR images show an IR temperature between -40 and -50 °C at the current aircraft position (black triangle) which indicates that the research aircraft was above, but close to cloud top at both occasions.

## 3.2  Trace gas measurements

Deep convective transport generally occurs in cumulonimbus systems accompanied by high cloud tops. Figure 3 shows the temporal development of the observed trace gases during MF12 and MF14. An overview of MF10 can be found in Figure S1 of the Supplement. Blue and green shadowed plot areas show the time intervals when the research aircraft had passed areas of high cloud tops as shown in Figure 2. The respective geographical positions of the aircraft are highlighted in Figure 1c and 1d. Indicators for deep convection from the marine boundary layer are enhanced concentrations of CO, DMS, $H_2O_2$ and $CH_3I$ and reduced $O_3$ at the flight altitude. In the absence of lightning we expect decreased concentrations of NO in convective areas due to the vertical transport of NO-poor marine boundary layer air. In contrast, we expect enhanced NO concentrations in the presence of lightning (Pollack et al., 2016; Ridley et al., 2004; Lange et al., 2001; DeCaria et al., 2000; Chameides et al., 1987). For MF12, $O_3$ was low while passing the area of enhanced cloud tops after take-off with an average of $41\pm2\,\mathrm{ppb}_v$ (10:30 - 11:45 UTC). The IR satellite image subsequently shows lower cloud tops and the measured $O_3$ concentrations at the same flight altitude of 12.9 km was on average $69\pm15\,\mathrm{ppb}_v$ (11:45 - 12:55 UTC). The same area of high cloud tops was passed on the way back at a higher flight altitude of 14.4 km. Before entering the area, $O_3$ was on average $76\pm5\,\mathrm{ppb}_v$ (16:15 - 17:00 UTC) and then decreased to $46\pm11\,\mathrm{ppb}_v$ (17:05 - 18:15 UTC). For MF14, the research aircraft also passed an area of elevated cloud tops after take-off (11:20 - 12:05 UTC) and before landing (17:50 - 18:25 UTC) with $O_3$ average concentrations of $34\pm2$ and $36\pm2\,\mathrm{ppb}_v$, respectively, at a flight altitude of 13.2 km. $O_3$ concentrations measured between these areas were higher by around 30 % with $48\pm10\,\mathrm{ppb}_v$ at an altitude of $12.6\pm0.3$ km. At similar altitudes, MF10 showed $O_3$ concentrations of $72\pm6\,\mathrm{ppb}_v$. Besides the observed convective influence from the $O_3$ measurements, concentrations were overall larger for MF10 and MF12 compared to MF14. This is likely a response to NO concentrations which influence $O_3$ production as discussed further below.

For MF12, DMS was significantly enhanced when passing the area of high cloud tops in the morning with a maximum value of 33 $\mathrm{ppt}_v$ and varying between 0 and 18 $\mathrm{ppt}_v$ at the same flight altitude outside this area. No DMS was detected when passing the convective area in the evening which is possibly due to the higher altitude of the research aircraft compared to the morning hours. The IR cloud top images in Figure 2a show that the aircraft was likely above the cloud top while the convective influence is highest within the cloud. DMS concentrations during MF14 were on average $27\pm17\,\mathrm{ppt}_v$ and $14\pm9\,\mathrm{ppt}_v$ passing the first and the second high cloud top area, respectively, and below the detection limit in between which clearly shows the effect of convection from the marine boundary layer. A maximum value of 58 $\mathrm{ppt}_v$ was reached passing the first cloud top area which was higher compared to the MF12 maximum. For MF10, DMS concentrations were below the detection limit at comparable altitudes for the whole flight. $CH_3I$ and $H_2O_2$, too, reached maximum concentrations when passing high cloud top areas during MF12 and MF14 and lower values at similar altitudes with lower cloud tops. NO concentrations were $169\pm85\,\mathrm{ppt}_v$ for MF10 at $13.6\pm0.7$ km. For the identified convective areas during MF14, NO was close to zero (0 - 37 $\mathrm{ppt}_v$), and slightly enhanced in

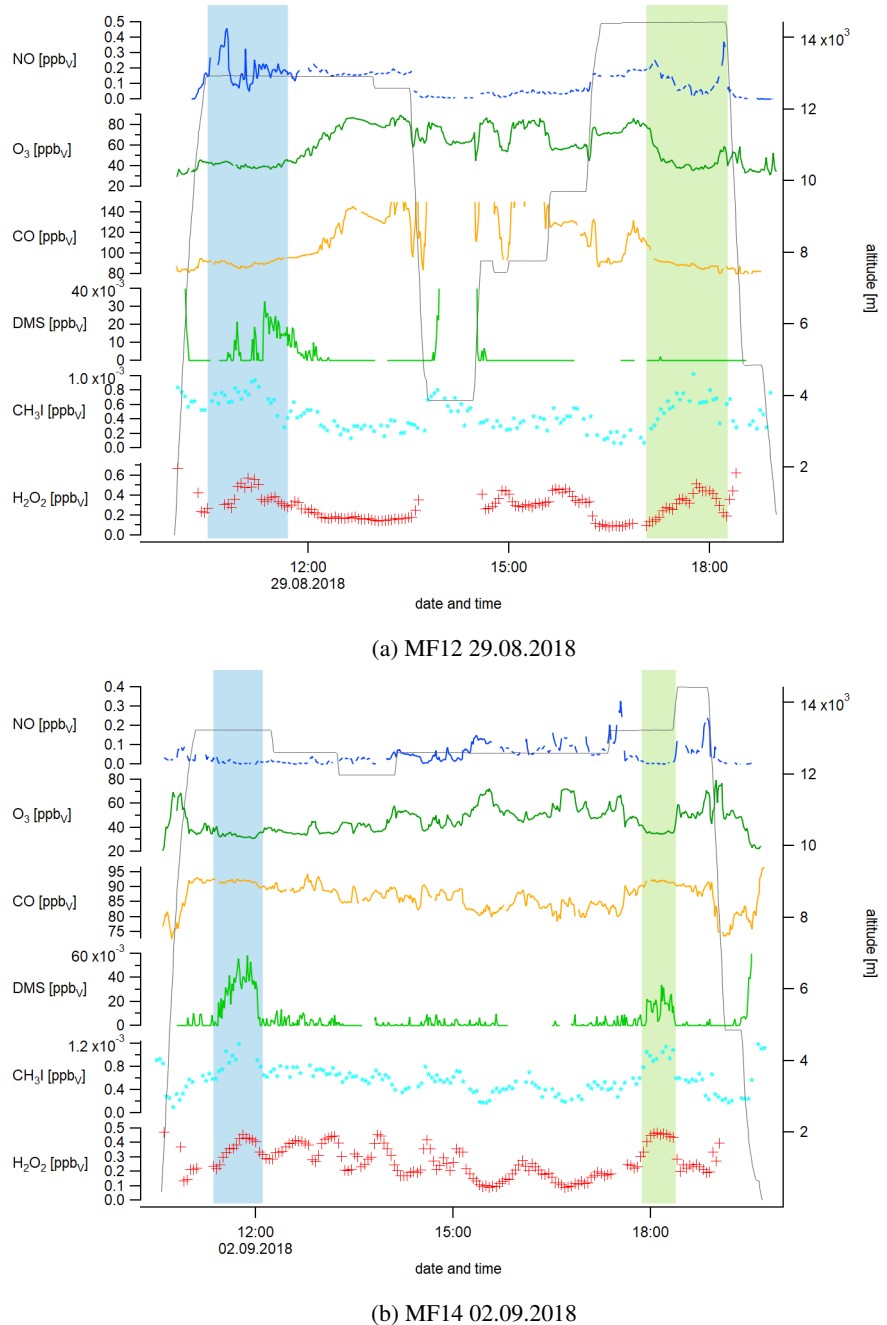

(a) MF12 29.08.2018

(b) MF14 02.09.2018

**Figure 3.** Overview of the temporal development of the observed trace gases NO, $O_3$, CO, DMS, $CH_3I$ and $H_2O_2$ during measurement flights MF12 and MF14. Blue and green bars show the time intervals for which the research aircraft had passed high cloud tops (compare Figure 1c and 1d).The overview for MF10 can be found in Figure S1 of the Supplement.

between with an average value of $56\pm50\,\mathrm{ppt}_v$. These observations demonstrate the occurrence of convection and the absence of lightning. In contrast, MF12 showed clear signs of lightning, particularly when passing the identified convective area in the morning. The measured NO concentrations showed characteristic spikes with a maximum of $459\,\mathrm{ppt}_v$ which is by more than one order of magnitude higher than the detected signals during MF14. NO concentrations for MF12 outside the convective areas were $169\pm21\,\mathrm{ppt}_v$ which is very similar to NO levels during MF10. Backward trajectories for MF10 and MF12 (outside of convection) (Figure S2 of the Supplement) show that the air originated from the African continent where lightning is frequent. Together with the absence of large spikes as observed for MF12 in the area of high cloud tops - an indicator for fresh lightning - the increased background level of NO for MF10 and MF12 was likely due to aged nitric oxide from thunderstorm activity over the ITCZ, mainly West Africa. Possible explanations for the lower NO background concentration during MF14 could be that the flight path was further away from coastal Africa and that the flight data were influenced by convective processes with upward transport of low NO air from the marine boundary layer. Assuming dominant $NO_x$-limited chemistry as suggested by Tadic et al. (2021), low NO background levels induce low $O_3$ levels. In the convective areas during MF14 with particular low NO concentrations, an $O_3$ destruction regime might have been present. In contrast, high NO background levels during MF10 and MF12 lead to the observed high $O_3$ background as mentioned earlier. CO background levels for MF10 were $84\pm7\,\mathrm{ppb}_v$. For MF14, CO was enhanced when passing high cloud tops with $92\pm1\,\mathrm{ppb}_v$, and lower in between these areas with $86\pm3\,\mathrm{ppb}_v$ which emphasizes the updraft of CO-rich air from the earth's surface. CO concentrations in the high cloud top areas for MF12 were comparable to those for MF14, but significantly lower compared to adjacent areas with low cloud tops observed in the southern hemisphere. From these observations it seems that the inflow to the thunderstorms was in the northern hemisphere, while the background mixing ratios of CO were higher in the southern hemisphere due to biomass burning.

Figures S3 and S4 of the Supplement show the flight track of MF12 and MF14, respectively, color-coded according to the measured trace gas concentrations which underlines the geographic allocation of the convective areas.

## 3.3 Deep convection

Figure 4 shows the vertical concentration profiles of atmospheric trace gases averaged for all CAFE Africa take-offs and landings on Cape Verde in gray. Flights before sunrise and after sunset were excluded for NO, $O_3$, CO, $H_2O_2$ and DMS. Figure S5a of the Supplement shows an overview of all data points that were included. Generally, for take-off and landing the data points before reaching and after leaving a constant flight altitude, respectively, were considered. $CH_3I$ data were available for MF11, MF12, MF14 and MF15. Each data point in the vertical profile is the average of all data measured at this altitude $\pm\,500\,\mathrm{m}$ providing a background profile of atmospheric trace gases around Cape Verde. Please note that these profiles represent background conditions in the northern hemisphere. Southern hemisphere profiles generally show higher mixing ratios for CO and $O_3$ due to biomass burning over south Africa and throughout the hemisphere (Figure S6 of the Supplement). Red colors represent MF14 and green colors show MF12. Filled symbols represent areas with convection and open symbols represent areas without convection according to the results from Sect. 3.2. For filled symbols, we differentiate between circles for the first passing of a convective area and triangles for the second passing. An overview of the symbols representing certain

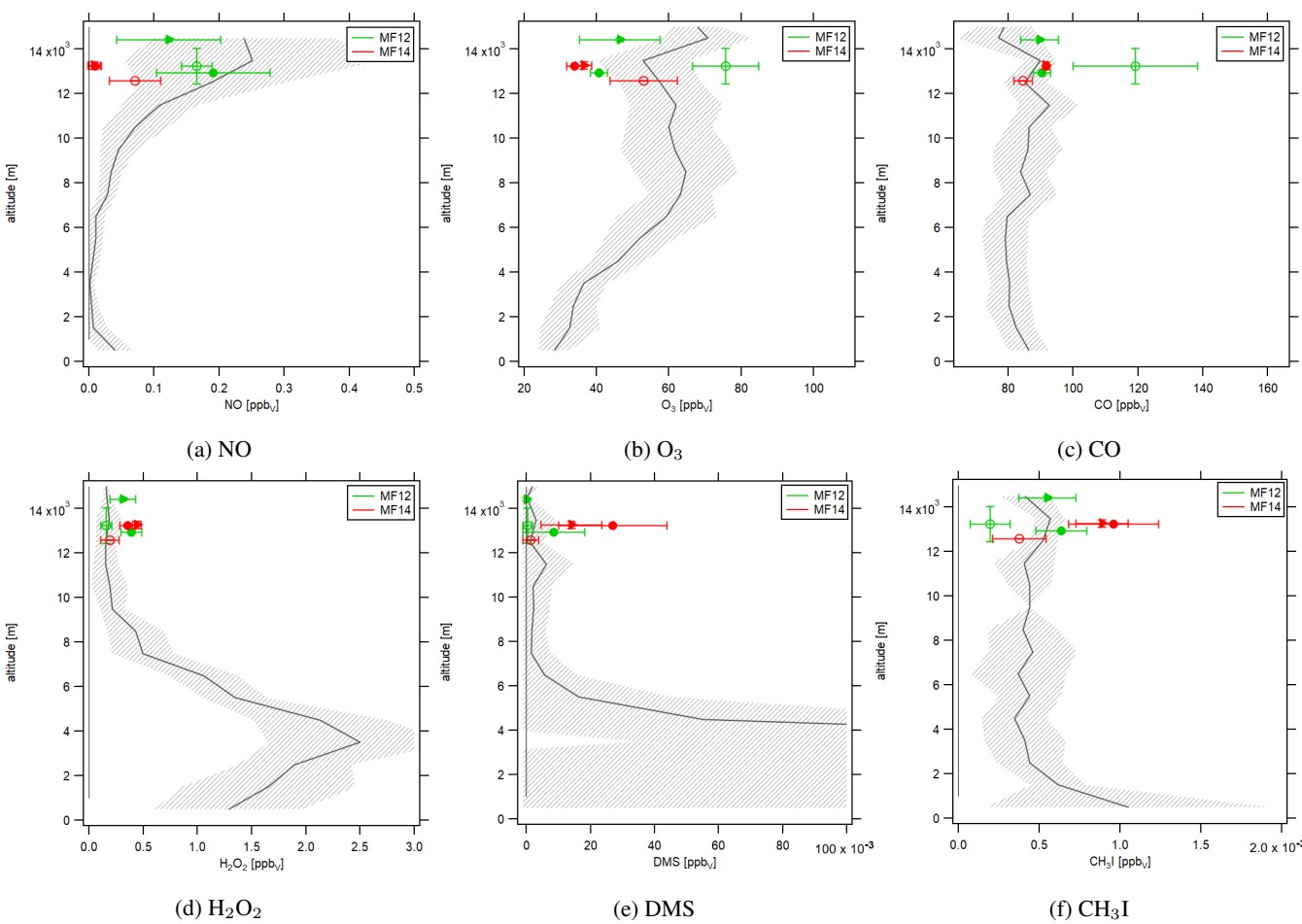

**Figure 4.** Vertical profiles of the background trace gas concentrations during CAFE Africa around Cape Verde (gray) and the average trace gas concentrations for convective (filled symbols) and non-convective (open symbols) areas during MF12 (green) and MF14 (red).

flight sections can be found in Figure S5b of the Supplement. Figure 4a shows NO concentrations which are lowest at low altitudes and increase with height. Ground-level concentrations were slightly enhanced due to airport emissions but can be assumed negligible at ground-level altitudes over the ocean surface. The large enhancement and increased variability of NO

at altitudes above 10 km was due to the overall effect of lightning associated with the position of the ITCZ just south of the Cape Verde islands (Figure 2 of Tadic et al. (2021)). Average NO concentrations for MF14 in convective areas were close to zero - emphasizing the vertical updraft of NO-poor air from the marine boundary layer - while NO in the non-convective area was enhanced. All data points for MF12 are within the variability range of the background profile which is what we expect for non-convective areas. For the MF12 convective areas, two opposing trends appear which are the vertical transport of NO-poor

air from ground level altitudes and the generation of fresh NO at high altitudes through lightning. From Figure 3a we suggested the occurrence of fresh lightning primarily for the early passing of the convective area. This is in accordance with the green

circle (first passing) being situated at higher NO levels compared to the green triangle (second passing). $O_3$ concentrations are shown in Figure 4b. Ground-level $O_3$ was low, increasing with altitude up to 8 km, and reaching a concentration of $61\pm6$ ppb$_v$ aloft. Average concentrations in convective areas were reduced for both MF12 and MF14 while they were enhanced in non-convective areas. Low altitude CO concentrations were enhanced through ocean emissions and transport from the continent, and led to a slight increase in concentrations in convective areas. As described above, the non-convective area of MF12 was heavily influenced by a biomass burning plume. The $H_2O_2$ background profile shown in Figure 4d peaks at around 3 - 4 km altitude where the NO profile is lowest. NO-poor air induces an $O_3$ destructive regime, enhancing the abundance of peroxyradicals forming $H_2O_2$. Ground-level $H_2O_2$ was lower due to surface uptake. Non-convective areas of MF12 and MF14 are well represented by the $H_2O_2$ background profile while concentrations in convective areas were enhanced. DMS and $CH_3I$ (Figures 4e and 4f) were elevated at low altitudes due to ocean emissions and possibly dust emissions and decreased with altitude. Again, concentrations of trace gases in convective areas showed an enhancement compared to non-convective areas. Figure S7 of the Supplement presents the background profiles including average concentrations of MF10 and MF14. As expected, values for MF10 are well described by the background profiles. For the trace gases CO, $H_2O_2$ and DMS, open symbols (representing non-convective average concentrations) for MF10 and MF14 are situated very close to the background profile. Filled symbols for MF14 are enhanced, which corroborates the effect of convection. For $O_3$ and NO, convective average concentrations were significantly reduced while again, open symbols fall within the variability range of the background profile. NO concentrations for MF14 were slightly lower compared to MF10 for non-convective areas due to the reasons discussed above.

While the discussed trace gases usually have a relatively short lifetime of the order of days, it is also possible that convection has occurred in a different location and the trace gases were transported to the point of observation through advection. Backward trajectories can be used to examine this hypothesis. Figure S8 and S9 of the Supplement show the color enhanced infrared satellite images of MF14 including the flight track and the backward trajectories for the prior five days. Black crosses mark the location of each calculated "air parcel" on its trajectory at the time when the satellite image was taken. It is shown that the backward trajectories are crossing the convective clouds of the developing cyclone several times. In contrast, Figure S10 of the Supplement shows the analogous images for a section of the flight track further west where convection was not observed. The calculated "air parcels" on the backward trajectories were ahead of the developing cyclone at all times and never passed a convective system.

### 3.4 Lightning

In Sect. 3.1, 3.2 and 3.3, we have presented evidence on the occurrence of deep convection during MF12 and MF14. We have hypothesized from the observed NO concentrations in the convective areas that the low-pressure system observed during MF12 included lightning while the tropical storm observed during MF14 did not show any lightning activity. Figure 5 shows the lightning strokes as asterisks and the flight tracks as lines for MF10 (Figure 5a), MF12 (Figure 5b) and MF14 (Figure 5c) color-coded according to the time-of-day obtained from the WWLLN. The WWLLN provides real-time lightning data covering almost the entire globe including oceans and remote locations. This is accompanied by a lower detection efficiency compared to local networks. Several studies have investigated this topic suggesting a WWLLN global detection efficiency of

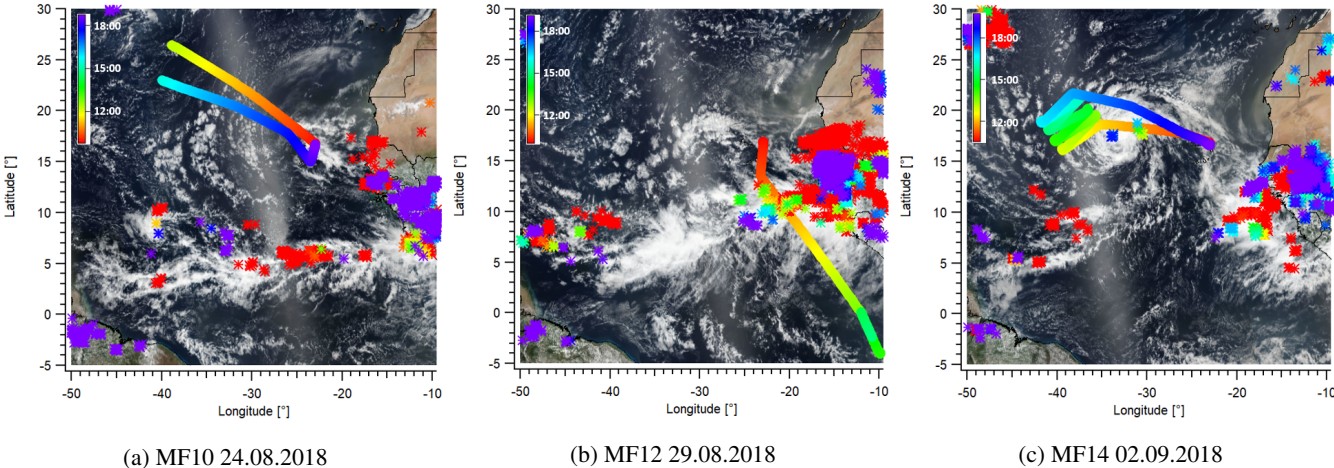

**Figure 5.** Lightning flashes shown as asterisks detected by the WWLLN on August 24, August 29 and September 2, 2018. Flashes and flight tracks are color-coded according to the time-of-day (Red shows morning hours and blue shows evening hours.) Background satellite images were obtained from the NASA Worldview application (NASA Worldview, 2020).

around 10 % with constant improvements through an increasing number of stations (Holzworth et al., 2019; Bürgesser, 2017; Virts et al., 2013; Abarca et al., 2010). Allen et al. (2019) calculated a detection efficiency of around 12 % for tropical Africa (LON -30° to 90°, LAT -30° to 30°) and of around 30 % for tropical America (LON -150° to -30°, LAT -30° to 30°) for 2011. At the same time, the WWLLN is capable of detecting almost any storm with lightning (Hutchins et al., 2012a; Jacobson et al., 2006). As expected, no lightning strokes were observed during MF10. For MF12, many lightning strokes were detected in the area with deep convection - many of which occurred in spatial and temporal proximity to the research aircraft. Figure 5b only shows the outbound flight part of MF12 as the inbound flight was stacked and much of lightning occurred between take-off and noon. Finally, for MF14, very few lightning strokes were detected by the WWLLN in the area of the tropical storm which is in accordance with the low observed NO concentrations.

## 4  Conclusions

In this study, we have presented in situ observations of a tropical cyclone which developed into hurricane Florence during the Atlantic hurricane season 2018. A nascent low-pressure system was observed during a measurement flight with the research aircraft HALO on August 29 and after a tropical storm had developed on September 2. We observed deep convection for both, the tropical wave and the tropical storm, based on in situ observations, supported by color-enhanced infrared imagery taken by the satellite GOES-16. Measured NO concentrations suggest significant occurrence of lightning only in the tropical wave, but not in the tropical storm. This hypothesis is confirmed by the lightning strokes detected through the WWLLN. Our result is consistent with previous studies for example by DeMaria et al. (2012), Zhang et al. (2015) and Abarca et al. (2011) who found decreasing lightning activities with increasing cyclone strength. While these studies are based on satellite and ground based

observations, we present the first in situ observations in support of this hypothesis demonstrating that convective injection of marine boundary layer air can occur without NO production from lightning. In future, more in situ observations of deep convection and lightning activity in tropical cyclones with variating strength should be performed and reported in order to consolidate and expand the present knowledge of lightning in deep convective systems and its role in atmospheric chemistry.

*Data availability.* Data measured during the flight campaign CAFE Africa are available to all scientists agreeing to the CAFE Africa data protocol. Lightning data are available upon request from the World Wide Lightning Location Network.

*Author contributions.* HF had the idea. CMN and HF designed the study. CMN analyzed the data and wrote the manuscript. IT measured and provided the NO and CO data. DD measured and provided the $H_2O_2$ data. JW, AE, NW and LE measured and provided the DMS and $CH_3I$ data. $O_3$ data were received from FO. HH and IGA calculated the backward trajectories. HF, JL, HH and JW significantly contributed to planning and operating the research campaign.

*Competing interests.* The authors have no competing interests to declare.

*Acknowledgements.* We would like to thank Uwe Parchatka for his assistance with the measurement of NO, Bettina Hottmann for her support of the measurement of $H_2O_2$ and Efstratios Bourtsoukidis for his assistance with the measurement of methyl iodide during CAFE Africa. We acknowledge the collaboration with the DLR (German Aerospace Center) during CAFE Africa. We acknowledge the Cooperative Institute for Research in the Atmosphere, Colorado State University, and NOAA's Center for Satellite Research, Fort Collins Colorado for using imagery from the TC-Realtime web page. We acknowledge the Naval Research Laboratory for providing IR satellite images on the NRL Tropical Cyclone Page. The authors wish to thank the World Wide Lightning Location Network (http://wwlln.net), a collaboration among over 50 universities and institutions, for providing the lightning location data used in this paper. We acknowledge the use of imagery from the NASA Worldview application (https://worldview.earthdata.nasa.gov), part of the NASA Earth Observing System Data and Information System (EOSDIS). This work was supported by the Max Planck Graduate Center with the Johannes Gutenberg-Universität Mainz (MPGC).

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
