# Peer review of "Measurement report: In situ observations of deep convection without lightning during the tropical cyclone Florence 2018."

_Atmospheric Chemistry and Physics, 2021_

## Author Comment (AC1)

**Referee 1:**

We would like to thank the referee for taking the time to review our manuscript and the valuable feedback. We have corrected our manuscript according to the referee's comments and think it is now significantly improved.

*This manuscript describes the atmospheric chemistry observations from the HALO aircraft during flights into a tropical storm and a tropical wave during the CAFE-Africa field mission. The tropical wave contained lightning and as a result significant enhancement of NO was noted. However, the tropical storm contained little or no lightning, and NO was not enhanced. Other chemical species that were considered include CO, O3, DMS, CH3I, and H2O2. The findings for all of these species were as expected, with enhancements of CO, DMS, CH3I, and H2O2 noted in the air parcels affected by deep convective transport from the marine boundary layer. O3 minima were also noted in the upper troposphere resulting from convective transport of low O3 boundary layer air.*

*The authors claim that this is the first report of in-situ chemical observations in deep convection in tropical cyclones with and without lightning. This is not entirely true. The authors need to reference the following papers and discuss their results in relation to them:*

*Newell, R., et al., (1996) Atmospheric sampling of Supertyphoon Mireille with NASA*

*DC-8 aircraft on September 27, 1991, during PEM-West A, J. Geophys. Res., 101, 1853-1871.*

*Roux, F., et al. (2020) The influence of typhoons on atmospheric composition deduced*

*from IAGOS measurements over Taipei, Atmos Chem. Phys., 20, 3945–3963, 2020*

*https://doi.org/10.5194/acp-20-3945-2020.*

*The Newell results show NO enhancements due to lightning (discussed further by Davis et al., 1996) in some portions of the storm (near eye wall), but not throughout the storm system. However, the observations reported by Roux et al. do not include NO, but should also be referenced.*

*The authors need to modify this claim in both the introduction and summary.*

We would like to thank the referee for this comment and the suggested papers. However, we politely disagree with the referee about the lack in novelty of our work. Newell et al. describes in-situ measurements of trace gases in different parts of the Typhoon Mireille, 1991. They found indicators for convective uptake from the boundary layer to the upper troposphere which was strongest in the wall cloud area. The authors suggest lightning as one source of increased NO concentrations by in-situ measurements. The resulting photochemistry is further discussed by Davis et al. In contrast, we compare in-situ measurements in a tropical storm and a weaker tropical wave. Our central statement is that both systems developed deep convection but only the weaker tropical wave had lightning which we show by in-situ measurements. Roux et al. also reports evidence on convective transport by in-situ measurements, but the absence of NO measurements does not allow for the in-situ detection of lightning. Nevertheless, we agree with the referee that the suggested papers are very valuable in regards to in-situ aircraft

measurement of convective uplift from the boundary layer in tropical cyclones and have added text to our manuscript.

Lines 89 ff.: Some studies have investigated trace gas concentrations and convective uplift in the upper troposphere through aircraft observations. Newell et al. (1996) reported in-situ observations of deep convection in the Typhoon Mireille in 1991 which they found to be strongest in the wall cloud region. Roux et al. (2020) found the convective uplift of boundary layer air as well as the inflow of lower stratospheric air to the upper troposphere based on measurements of CO, $O_3$ and $H_2O$ during aircraft typhoon observations over Taipei in 2016. In contrast, studies of lightning activity within convective systems over the ocean and in tropical cyclones are predominantly based on satellite data and ground-based observations from the WWLLN (University of Washington; Abreu et al., 2010; Bürgesser, 2017; Hutchins et al., 2012b; Bucsela et al., 2019). Generally, data from in situ chemical measurements in the upper troposphere are sparse and to our knowledge, the in situ aircraft observation of deep convection in tropical cyclones accompanied by and in the absence of lightning depending on the stage of development has not been reported before.

*Detailed Comments:*

*Lines 12-13: The ITCZ is not a broad area spanning +/- 20 degrees from the equator. It most often lies within this belt, but the convection associated with the ITCZ covers a much smaller range of latitude. Tropical cyclones may develop from ITCZ convection, but often they are not associated with the ITCZ.*

Thank you. We have changed this sentence emphasizing that this band is not equal to but does include the ITCZ.

Lines 12 f.: Tropical cyclones are low-pressure systems evolving over warm tropical waters usually close to the equator (± 20°) - an area which includes the so-called Intertropical Convergence Zone (ITCZ).

*line 17: 15 km or higher*

We have corrected this.

Lines 16 ff.: In this region of high ocean temperature and intense solar radiation humid air can rise deeply into the troposphere up to 15 km and higher (Collier and Hughes, 2011; Deutscher Wetterdienst).

*line 31: Deep Convective Clouds and Chemistry*

Corrected.

Lines 35 f.: (…) the DC3 (Deep Convective Clouds and Chemistry) field campaign. the DC3 (Deep Convective Clouds and Chemistry) field campaign.

*line 33: ....downwards due to gravity in the presence of supercooled water.....*

Thank you, we have included this.

Lines 38 ff.: Collision of light ice particles moving upwards in cumulonimbus clouds and graupel particles moving downwards due to gravity in the presence of supercooled water

induces electric charge separation which accumulates and discharges spontaneously as a lightning flash.

*line 42:  Add reference to Cecil et al. (2014, Atmos. Res.)*

We have added the suggested reference.

Lines 47 f.: These results are in line with other published works e.g. Cecil et al. (2014), Xu and Zipser (2012) and Xu et al. (2010).

*line 78:  WWLLN:  there are many journal references to WWLLN that are available to use here in addition to the website.  Please include some of them.*

Thank you for pointing this out. We have added a selection of references regarding the WWLLN.

Lines 94 f.: (…) and ground-based observations from the WWLLN (University of Washington; Abreu et al., 2010; Bürgesser, 2017; Hutchins et al., 2012b; Bucsela et al., 2019).

*line 78-79:  Data from in situ chemical measurements in the upper troposphere are sparse.*

We have changed this.

Lines 95 ff.: Generally, data from in situ chemical measurements in the upper troposphere are sparse (…).

*line 83:  ...evidence of the chemical impacts of deep convection....*

Changed.

Line 100 f.: The data are examined for evidence of the chemical impacts of deep convection and lightning activity.

*Figure 2:  It would be helpful to also show the 1200 UTC satellite image for MF12*

We agree with the referee that the UTC 12:00 satellite image for MF12 would be useful in this analysis, but unfortunately no satellite images are available before UTC 18:00. We have added a note regarding the availability in the caption of Figure 2.

Flight tracks with color enhanced infrared imagery obtained from the Naval Research Laboratory Tropical Cyclone page (Naval Research Laboratory, 2020) for MF12 and MF14 (no image availability before MF12 UTC 18:00).

*line 125:  Deep convective transport generally occurs....*

*Corrected.*

Line 148: Deep convective transport generally occurs in cumulonimbus systems accompanied by high cloud tops.

*line 132: There are many other references for enhanced NO from lightning, and some of them should be given here. Examples:*

*Chameides et al., 1987 - JGR; Ridley et al., 1987 - JGR; DeCaria et al. (2000) - JGR; Ridley et al. (2004) - JGR; Pollack et al. (2016) – JGR*

Thank you for these suggestions which we have added to our manuscript.

Lines 154 f.: In contrast, we expect enhanced NO concentrations in the presence of lightning (Pollack et al., 2016; Ridley et al., 2004; Lange et al., 2001; DeCaria et al., 2000; Chameides et al., 1987).

*lines 140-141: ....concentrations wre overall larger for MF10 and MF12 than for MF14.... This makes more sense with regard to the following sentence.*

We have changed that.

Lines 163 ff.: Besides the observed convective influence from the $O_3$ measurements, concentrations were overall larger for MF10 and MF12 compared to MF14.

*line 145: ...was likely above... Line 121 says the aircraft was at similar altitude as cloud tops.*

Thank you, we have corrected this. The aircraft is at a similar altitude as the cloud top in the morning and likely above the cloud top in the afternoon.

Lines 140 ff.: The flight altitude for MF12 at 18:00 UTC shown in Figure 2a was 14.4 km while overpassing an area of elevated clouds. The according temperature was - 68±1 °C, so the aircraft was likely above the cloud top. For MF12, the same area was passed in the morning at an altitude of 12.9 km, but no IR image was available. The temperature was - 56±1 °C. Assuming a similar cloud elevation in the morning, the research aircraft was at a similar altitude as the cloud top.

*line 147: mention that the peak DMS on MF14 was ~50 ppbv, which is greater than on MF12*

We have added this point to section 3.2.

Lines 172 f.: A maximum value of 58 ppt$_v$ was reached passing the first cloud top area which is higher compared to the MF12 maximum.

*line 157: ....for MF10 and MF12 (outside of convection)....*

Corrected.

Lines 181 f.: Backward trajectories for MF10 and MF12 (outside of convection) (Figure S2 of the Supplement) show that the air originated from the African continent where lightning is frequent.

*lines 161-162: ...convective processes with upward transport of low NO air from the marine BL*

Thank you for the suggestion, we have added this.

Lines 186 f.: (…) the flight data were influenced by convective processes with upward transport of low NO air from the marine boundary layer.

*line 189:  ....ITCZ just south of the Cape Verde Islands.    Lines 159-160 say it is due to lightning over West Africa.*

The ITCZ is an area of enhanced lightning activity which is highest above continental areas and - at the latitudes we are looking at - predominantly West Africa. We have added text to clarify this.

Lines 184 f.: (…) the increased background level of NO for MF10 and MF12 was likely due to aged nitric oxide from thunderstorm activity over the ITCZ, mainly West Africa.

*line 227:  Need to mention that WWLLN is only detecting some small fraction of the total lightning that occurred because it has a rather low detection efficiency.  Need to reference papers on WWLLN detection efficiency.  See the WWLLN webpage.*

We would like to thank the referee for noting this. We have added a section describing the WWLLN detection efficiency.

Lines 252 ff.: The WWLLN provides real-time lightning data covering almost the entire globe including oceans and remote locations. This is accompanied by a lower detection efficiency compared to local networks. Several studies have investigated this topic suggesting a WWLLN global detection efficiency of around 10% with constant improvements through an increasing number of stations (Holzworth et al., 2019; Bürgesser, 2017; Virts et al., 2013; Abarca et al., 2010). Allen et al. (2019) calculated a detection efficiency of around 12% for tropical Africa (LON -30° to 90°, LAT -30° to 30°) and of around 30% for tropical America (LON -150° to -30°, LAT -30° to 30°) for 2011. At the same time, the WWLLN is capable of detecting almost any storm with lightning (Hutchins et al., 2012a; Jacobson et al., 2006).

---

## Author Comment (AC2)

**Referee 2:**

We would like to thank the referee for taking the time to review our manuscript and the valuable feedback. We have corrected our manuscript according to the referee's comments and think it is now significantly improved.

*The manuscript "Measurement report: In situ observations of deep convection without lightning during the tropical cyclone Florence 2018." by Nussbaumer et al. describes HALO aircraft observations (3 flights during CAFE campaign) of atmospheric trace gases in a tropical storm and discusses the evidence of (or the lack of) lightning occurrence during this event. The manuscript, while very synthetic and somewhat lacking detailed discussion, is very well written, clear and, in my opinion, useful and pertinent for the ACP readers. The lack of deep discussion might be due to the fact that this is a "Measurement Report" manuscript type. I wonder if, with a bit more committment in drawing conclusions, this might as well be a "Letter"/"Short Communication" manuscript. I leave these considerations up to the Editor and Authors. In any case, I recomment this paper for publication in ACP after these minor issues are clarified.*

*1) I agree with the other Reviewer that references are lacking for other previous aircraft campaigns. I add, for the observation of the UTLS composition in deep convection area, the StratoClim campaign; maybe this reference is a good pick, with respect to StratoClim:*
*Bucci et al.: Deep-convective influence on the upper troposphere–lower stratosphere composition in the Asian monsoon anticyclone region: 2017 StratoClim campaign results, Atmos. Chem. Phys., 20, 12193–12210, https://doi.org/10.5194/acp-20-12193-2020, 2020.*

We have extended our literature discussion of previous aircraft campaigns regarding (1) deep convection where we have included the referee's literature suggestion and (2) in-situ observations in tropical cyclones.

(1) Lines 33 ff.: Deep convection can affect trace gas concentrations in the upper troposphere which was for example shown by Dickerson et al. (1987) who reported increased concentrations of NO, CO, O3 and other reactive species in a thunderstorm outflow over the Midwestern United Stated in 1985 and Barth et al. (2015), the latter based on observations during the DC3 (Deep Convective Clouds and Chemistry) field campaign. Similar observations were made by Bucci et al. (2020) who reported convective uplift in the upper troposphere / lower stratosphere based on the StratoClim aircraft campaign over Nepal in 2017.

(2) Lines 89 ff.: Some studies have investigated trace gas concentrations and convective uplift in the upper troposphere through aircraft observations. Newell et al. (1996) reported in-situ observations of deep convection in the Typhoon Mireille in 1991 which they found to be strongest in the wall cloud region. Roux et al. (2020) found the convective uplift of boundary layer air as well as the inflow of lower stratospheric air to the upper troposphere based on measurements of CO, $O_3$ and $H_2O$ during aircraft typhoon observations over Taipei in 2016. In contrast, studies of lightning activity within convective systems over the ocean and in tropical cyclones are predominantly based on satellite data and ground-based observations from the WWLLN (University of Washington; Abreu et al., 2010; Bürgesser, 2017; Hutchins et al., 2012b; Bucsela et al., 2019). Generally, data from in situ chemical measurements in the upper troposphere are sparse and to our knowledge, the in situ aircraft observation of deep convection in

tropical cyclones accompanied by and in the absence of lightning depending on the stage of development has not been reported before.

*2) L20-21: Maybe add a sentence to very briefly explain mechanisms of formation of tropical cyclones from tropical disturbances*

We have added text on the formation of tropical cyclones from tropical disturbances which is not fully understood today.

Lines 22 ff.: Wu et al. suggested that simultaneous occurring of convection and vorticity in disturbances favors tropical cyclone formation. However, the exact formation mechanism of tropical cyclones from tropical waves is not fully understood today (Wu and Takahashi), 2019; Frank and Roundy, 2006).

*3) L24: "...within about 5° of the equator..."*

We have corrected this.

Lines 26 f.: (…) while rotating systems do not develop within around 5° of the equator (…).

*4) L37: add the year of the publication "Zipser" in the text*

We have added the year (1994) of the Zipser publication in the text.

Line 43: Zipser (1994) reported significantly reduced lightning activity (…).

*5) L50-51: "Over the ocean...aircraft": to link with the previous sentence, you might probably very quickly cite NO source over land.*

We have added text and references on NO sources over land – with particular focus on West Africa.

Lines 58 ff.: NO sources over land are more versatile including anthropogenic emissions from industry, vehicles and biomass burning (partly natural) as well as natural sources from lightning and soil, the latter dominating over West Africa (Pacifico et al., 2019; Knippert et al., 2015).

*6) L58-60: "Another possible...iodide": you could add a few words on how methyl iodide is formed from dust*

We have added a short explanation of the mechanism of CH3I formation from dust and seawater as suggested by Williams et al., 2007 based on data from two field campaigns in Tenerife (MINATROC) and in the Tropical Atlantic (Ship campaign Meteor 55). Please note that the mechanism is not yet fully understood.

Lines 68 ff.: One possible explanation for the formation of CH3I is a substitution reaction of methoxy group containing species and iodide from seawater under the presence of iron ions from dust. However, the mechanism is not yet finally understood (Williams et al., 2007).

*7) L61-62: "Its lifetime depends on the abundance of OH and NO3 which oxidize DMS and ranges from 1 to 2 days" please rephrase (it sounds like "OH and NO3" or "DMS" range from 1 to 2 days...)*

We have rephrased the sentence.

Lines 72 f.: Its lifetime ranges from 1 to 2 days and depends on the atmospheric abundance of OH and $NO_3$ which oxidize DMS (Breider et al., 2010).

*8) L61: "abundance" --> "atmospheric abundance"*

We have changed this (see response to 7)).

*9) L78: "WWLLN": what is the meaning of this acronym?*

"WWLLN" is the acronym for World Wide Lightning Location Network. We decided to define the acronym when it is first used in line 49.

*10) L89: "...satellite images...": what are exactly these satellite images and from which instrument? NASA Worldview is just a data repository and visualisation tool but the exact type and origin of data should be mentioned.*

Thank you for noting this. We have added text about type and origin of the images.

Lines 108 ff.: The images were taken by the VIIRS (Visible Infrared Imaging Radiometer Suite) instrument carried by the NASA/NOAA satellite Suomi NPP (National Polar orbiting Partnership) based on a daily resolution (NASA Worldview, 2020).

*11) L92: "...compare Figure 1...": compare with what? You mean "...compare panels b and c of Figure 1..."?*

We meant to refer to Figure 1a for showing which parts of MF10 are in geographic proximity of MF14. We have clarified this in the text.

Lines 111 f.: We have restricted our analysis to data from MF10 which were obtained in these parts in a similar geographical area and altitude range as MF14 (as shown in Figure 1a).

*12) L116: "GEOS": please define acronym*

We have added the definition of the acronym "GEOS".

Lines 135 f.: The satellite images are colored according to the temperature deduced from IR emissions of cloud tops in °C as measured by the satellite GEOS-16 (Geostationary Operational Environmental Satellite).

*13) L123: "The colored IR images show that the research aircraft was above, but close to cloud top at both occasions.": It rather looks like you were flying under clouds tops (flight altitude in light green: ~-50°C, corresponding cloud top temperature in dark green: ~-60°C).*

We politely disagree with the reviewer as the current position of the research aircraft is marked with a black triangle. The cloud top color at the flight track of the respective current position is mainly blue suggesting an IR temperature between -40 and -50°C while the flight track is green: below -50°C. We have added text for clarification.

Lines 144 ff.: The flight altitude for MF14 at 12:00 UTC and at 18:00 UTC as shown in Figures 2b and 2c was 13.2 km and the temperature was - 58±1 °C. The colored IR images show an IR temperature between -40 and -50 °C at the current aircraft position (black triangle) which indicates that the research aircraft was above, but close to cloud top at both occasions.

*14) L127: "bars": is it "shadowed areas in the plot" or something like this?*

We have corrected this.

Lines 150 f.: Blue and green shadowed plot areas show the time intervals when the research aircraft had passed areas of high cloud tops as shown in Figure 2.

*15) Fig. 3: this is hardly visible, maybe this figure would be better organised as a 2 rows 1 column?*

We have increased the size of the subfigures according to the suggestion of the referee.